**Data Availability Statement:** All datasets and codes are available from GitHub: https://github.com/HYYANG9/BERT4FCA.

**Funding:** This study was funded by the Japan Society for the Promotion of Science (https://www.

# BERT4FCA: A method for bipartite link prediction using formal concept analysis and BERT

**Siqi Peng**[☯], **Hongyuan Yang**[iD][☯]*, **Akihiro Yamamoto**

Graduate School of Informatics, Kyoto University, Sakyo Ward, Kyoto City, Kyoto, Japan

☯ These authors contributed equally to this work.
* yang.hongyuan.67t@st.kyoto-u.ac.jp

## Abstract

Link prediction in bipartite networks finds practical applications in various domains, including friend recommendation in social networks and chemical reaction prediction in metabolic networks. Recent studies have highlighted the potential for link prediction by maximal bi-cliques, which is a structural feature within bipartite networks that can be extracted using formal concept analysis (FCA). Although previous FCA-based methods for bipartite link prediction have achieved good performance, they still have the problem that they cannot fully capture the information of maximal bi-cliques. To solve this problem, we propose a novel method for link prediction in bipartite networks, utilizing a BERT-like transformer encoder network to enhance the contribution of FCA to link prediction. Our method facilitates bipartite link prediction by learning more information from the maximal bi-cliques and their order relations extracted by FCA. Experimental results on five real-world bipartite networks demonstrate that our method outperforms previous FCA-based methods, a state-of-the-art Graph Neural Network(GNN)-based method, and classic methods such as matrix-factorization and node2vec.

## Introduction

*Bipartite link prediction* is the task of predicting the absence or presence of unobserved links in a *bipartite network* [1–3]. A bipartite network is a structure consisting of two disjoint sets of nodes and a set of edges where every edge only connects two nodes from different sets. Many real-world relational data can be naturally modeled as bipartite networks where the two sets of nodes represent two groups of *entities* and the edges represent their *links* or *relations* [4, 5]. In real-world bipartite networks, some links may be missing or have not been observed yet [1, 3]. This leads to the need to predict whether an unobserved link should be a potential new one and gives rise to the task of bipartite link prediction.

Currently, there are two types of bipartite link prediction. The first type focuses on predicting the relations between nodes from different node sets. For each pair of nodes from different node sets, it predicts whether there should be a link between them if they are not linked with an edge in the original network. For example, in a chemical-disease interaction database, for a

jsps.go.jp/english/e-grants/) in the form of a Grant-in-Aid for Scientific Research(B) to SP and AY [JP21H0499]. The funders had no role in study design, data collection and analysis, decision to publish, or preparation of the manuscript.

**Competing interests:** The authors have declared that no competing interests exist.

chemical and a disease whose interactions are unobserved, it predicts whether they should interact with each other. The second type focuses on predicting the relations between nodes from the same node sets. For two nodes from the same set that are not connected to the same node in the other set, it predicts whether there should be an unobserved node in the other set that is connected to both nodes. For example, in an author-paper network, for two authors that do not have a co-authorship, it predicts if they will have a new co-authorship in the future [6, 7]. Both types of bipartite link prediction have attracted increasing attention for high practical values [3, 8, 9]. Various methods have been proposed for both types of bipartite link prediction [2, 3, 10]. While some bipartite link prediction methods utilize external information, such as the properties of nodes [11], most methods make predictions based only on the structural features of the network. Notable methods that use only the structural features include pure rule-based methods like *Common Neighbors* [12] and node embedding methods such as *Node2Vec* [13] and autoencoder based embedding methods [14, 15].

Recently, research has found that the *bi-cliques* of a bipartite network represent important structural features and can be useful for bipartite link prediction [16–19]. A *bi-clique* is a complete sub-network of a bipartite network where every node in the first set is linked to every node in the second set. Bi-cliques represent clusters of strongly related entities. For example, in an author-paper network, a bi-clique represents a group of co-researchers and their publications. To capture the information of bi-cliques, research in [7] proposed *object2vec* and *attribute2vec*, which aims to embed the nodes of a bipartite network into a vector space based on their co-occurrence relationship in the maximal bi-cliques of the network. To achieve this, they used the method of *formal concept analysis* (FCA) [7, 20, 21]. FCA is a method for learning rules from a binary relational knowledge base, which is strongly connected with bipartite networks and bi-cliques. It aims at extracting and organizing *formal concepts* from a *formal context*. A *formal context* represents a collection of binary data of two disjoint sets of entities called *objects* and *attributes* and their binary relation. A *formal concept* represents a maximal group of objects that share the same set of attributes. If we regard the sets of objects and attributes in a formal context as the two node sets in a bipartite network, we will find that the binary relation in the formal context will be equivalent to the set of links in the bipartite network, and the formal concepts extracted from the formal context will be equivalent to the maximal bi-cliques extracted from the bipartite network. With such an equivalence, the co-occurrences of nodes in a maximal bi-clique can be converted to the co-occurrences of objects in a formal concept, which resembles the co-occurrences of words in a sentence. Thus, one can use the embedding models similar to the well-known word embedding model *Word2Vec* [22, 23] to embed the nodes into vectors [7]. Besides such a Word2Vec-like embedding method, in [20], the authors proposed another method for embedding the nodes based on bi-clique information with the help of *Bidirectional Long Short-Term Memory* (Bi-LSTM) [24]. Both methods have shown good performances [7, 20]. These two methods open up a novel strategy for bipartite link prediction, that is, to convert bipartite networks into formal contexts and use FCA to extract information on bi-cliques so that we can further process them with other methods. However, their methods have two limitations. First, they did not fully utilize all the information that FCA can extract. Second, their methods can only conduct the first type of link prediction, that is, predicting the relations between nodes from the same node set. We consider it better to design a general method that can conduct both types of bipartite link prediction.

To address these limitations, we propose a novel method called *BERT4FCA*, which is designed to capture more information provided by FCA and use such information to make both types of bipartite link prediction with the help of a popular method in natural language processing method called *BERT* or *Bidirectional Encoder Representations from Transformers* [24]. BERT is a method for training language models utilizing the *Transformer* architecture

[25]. It first pre-trains a large model on unlabeled free text to learn the co-relations between words and sentences; then, it fine-tunes the model on a small labeled dataset to fit a target downstream task. We chose BERT because we found that the information provided by FCA, such as the formal concepts and their hierarchical relations, shares some similarities with the input data that BERT takes. Hence, after processing the information provided by FCA and modifying the basic network structure of BERT, we will be able to capture all such information during pre-training. Then, by fine-tuning the pre-trained model to fit our target bipartite link prediction tasks, we are expected to get better results than those embedding methods thanks to the extra information we have captured. Additionally, in the field of natural language processing, while Word2Vec requires task-specific networks for downstream tasks, BERT can be fine-tuned for multiple tasks using the same pre-trained model, which is more convenient. We believe the convenience is preserved by our method since we do not need to design two different networks for two link prediction tasks.

Our main contributions are as summarized as follows:

- We propose a novel FCA-based method called BERT4FCA for bipartite link prediction, which can capture and learn more information extracted by FCA than previous FCA-based methods. We conduct ablation experiments to show that learning more information given by FCA contributes to higher link prediction performance.

- Experimental results show that our method outperforms all previous FCA-based methods as well as classic non-FCA-based methods on both types of bipartite link prediction.

- We demonstrate that the information on the order relation between maximal bi-cliques is beneficial for bipartite link prediction. To the best of our knowledge, no research has discovered it before.

- To the best of our knowledge, we are the first to provide a general method for using a BERT-like transformer architecture to learn information from the concept lattices provided by FCA. Although in this paper, we only discuss its application in bipartite link prediction, we plan to conduct further research on applying it to other FCA-related tasks.

The rest part of this paper is organized as follows. In the "Preliminaries" section, we start with some preliminaries, including bipartite networks, bi-cliques, FCA, and BERT. In the "Problem Formulation and Related Work" section, we present the problem formulation of bipartite link prediction and provide an overview of related work. In "The Proposed Method" section, we introduce and analyze our method BERT4FCA. In the "Experiment" section, we describe our experiments on five real-world datasets and discuss the results. Finally, in the "Conclusion and Future Work" section, we draw a conclusion and discuss our plans for future work.

## Preliminaries

### Bipartite networks and bi-cliques

**Bipartite networks.** A *bipartite network* $C$ is a triple $(U, V, E)$ where $U$, $V$ are two disjoint sets of *nodes* and $E$ is a set of *edges*. Each edge connects a node $u \in U$ to another node $v \in V$ and is denoted as $(u, v)$. That is, we should have $U \cap V = \emptyset$ and $E \subseteq U \times V$. Fig 1 gives an example of a bipartite network, where all nodes $u$ in $U$ are colored blue, and all nodes $v$ in $V$ are colored red. We can observe that edges only exist between pairs of nodes from different sets of nodes.

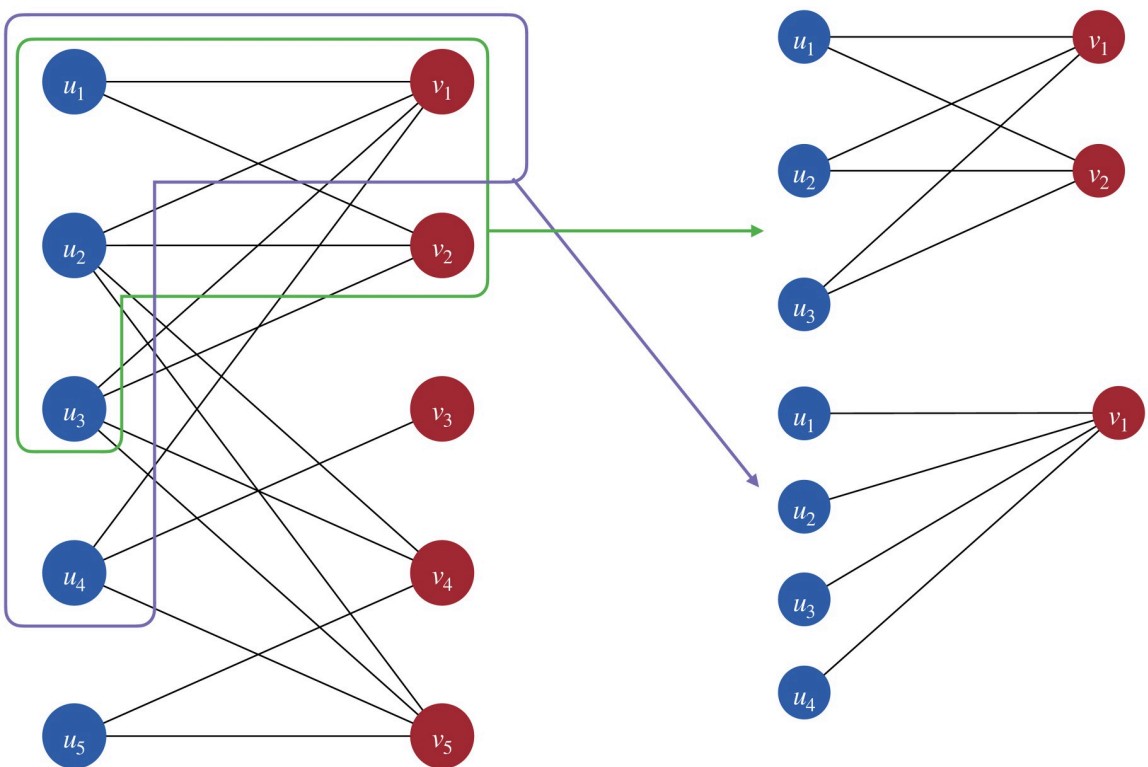

**Fig 1. An example of a bipartite network (*U*, *V*, *E*) and two of its bi-cliques.** The nodes in blue form the node set *U*, and the nodes in red form the node set *V*. The two sub-networks framed in green and purple are two maximal bi-cliques of the network.

**Sub-networks.** A bipartite network $C_1 = (U_1, V_1, E_1)$ is a *sub-network* of $C = (U, V, E)$ if both sets of nodes and the set of edges of $C_1$ are subsets of those of *C*, *i.e.*, $U_1 \subseteq U$, $V_1 \subseteq V$, and $E_1 \subseteq E$.

**Bi-cliques.** A bipartite network $C_1 = (U_1, V_1, E)$ is a *bi-clique* of $C = (U, V, E)$ if $C_1$ is a *sub-network* of *C* and there is an edge between every node pair from different subsets of nodes in $C_1$, *i.e.*, $U_1 \subseteq U$, $V_1 \subseteq V$, $E_1 \subseteq E$ and $U_1 \times V_1 = E_1$.

**Maximal bi-cliques.** A bi-clique $C_1 = (U_1, V_1, E_1)$ is a *maximal bi-clique* of a bipartite network $C = (U, V, E)$ if it is not a sub-network of any other bi-cliques of the bipartite network, *i.e.*, $\forall C_2 = (U_2, V_2, E_2)$ such that $U_1 \subseteq U_2 \subseteq U$ and $V_1 \subseteq V_2 \subseteq V$, $U_2 \times V_2 = E_2$ is satisfied if and only if $U_1 = U_2$, $V_1 = V_2$, and $E_1 = E_2$. Fig 1 gives some examples of maximal bi-cliques.

## Formal concept analysis (FCA)

*Formal concept analysis* (FCA) is a method for learning rules from a binary relational knowledge base called a *formal context*. Given a formal context, it aims to extract *formal concepts* and their hierarchical structures, which constitute *concept lattices*. In this paper, we only briefly introduce the minimal necessary notions of FCA. For a detailed introduction, please refer to [26–29].

**Formal contexts.** A formal context is a triple $\mathbb{K} := (G, M, I)$, where *G* is a set of *objects*, *M* is a set of *attributes*, and $I \subseteq G \times M$ is a binary relation called *incidence* that expresses which object has which attribute. We write *gIm* or $(g, m) \in I$ to express that the object $g \in G$ has the attribute $m \in M$.

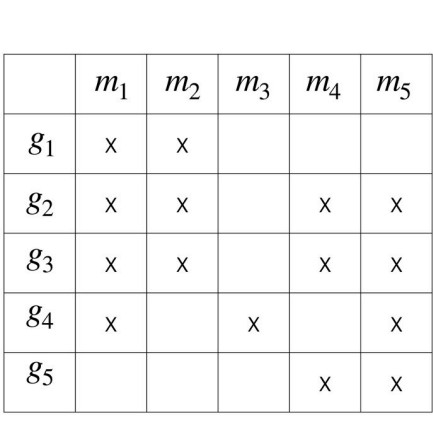

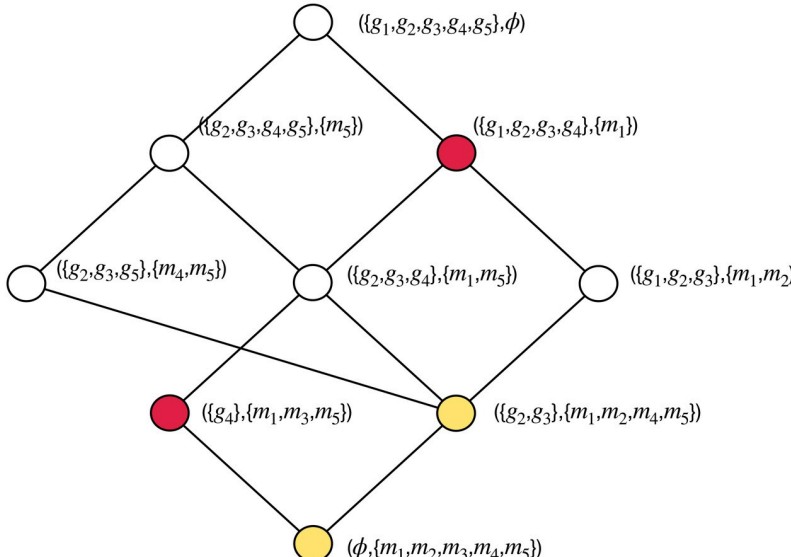

**Fig 2.** Left: A sample formal context. Right: The concept lattice corresponding to the formal context in the left panel. The nodes in yellow are neighbors and the nodes in red are not neighbors.

Formal contexts are illustrated in binary tables, as exemplified in the left of Fig 2, where rows correspond to objects and columns to attributes, and a cell is marked with a cross if the object in its row has the attribute in its column. In the context shown in the left panel of Fig 2, the marked cell represents that the object listed in the row possesses the corresponding attribute in the column.

**Formal concepts.** In a context $\mathbb{K} = (G, M, I)$, for subsets of objects and attributes $A \subseteq G$ and $B \subseteq M$, $(A, B)$ is called a *formal concept* if $\forall (A_1, B_1)$ such that $A \subseteq A_1 \subseteq G$ and $B \subseteq B_1 \subseteq M$, $A_1 \times B_1 \subseteq I$ is satisfied if and only if $A = A_1$ and $B = B_1$. If $(A, B)$ is a formal concept, $A$ is also called an *extent*, and $B$ is also called an *intent*.

**Concept lattices.** Given a context $\mathbb{K} = (G, M, I)$, the *concept lattice* of context $\mathbb{K}$, denoted by $\mathfrak{B}(\mathbb{K})$, is the structure that organizes the set of all concepts extracted from context $\mathbb{K}$ with the *hierarchical order* $<$. For two concepts $(A_1, B_1)$ and $(A_2, B_2)$, we write $(A_1, B_1) < (A_2, B_2)$ if $A_1 \subset A_2$ (which mutually implies $B_2 \subset B_1$).

Concept lattices are usually figuralized with line diagrams. For example, the line diagram shown in the right panel of Fig 2 represents the concept lattice of the context represented in the left panel of the same figure. In the diagram, nodes represent formal concepts and lines represent hierarchical orders.

**Neighbor of concepts.** In a concept lattice $\mathfrak{B}(\mathbb{K})$, for two concepts $(A_1, B_1)$ and $(A_2, B_2)$ such that $(A_1, B_1) < (A_2, B_2)$, $(A_1, B_1)$ is defined to be the *lower neighbor* of $(A_2, B_2)$ if $\nexists (A_3, B_3) \in \mathfrak{B}(\mathbb{K})$ such that $(A_1, B_1) < (A_3, B_3) < (A_2, B_2)$ [26]. In this case, $(A_2, B_2)$ is dually called the *upper neighbor* of $(A_1, B_1)$ or we may also directly say that $(A_1, B_1)$ and $(A_2, B_2)$ are *neighbors* or that they have the *neighboring relation*.

In the line diagram representation of a concept lattice, if two concepts are neighbors, they will be shown as two nodes directly connected by a line. For example, in the concept lattice shown on the right of Fig 2, the nodes in yellow are neighbors, and the nodes in red are not neighbors.

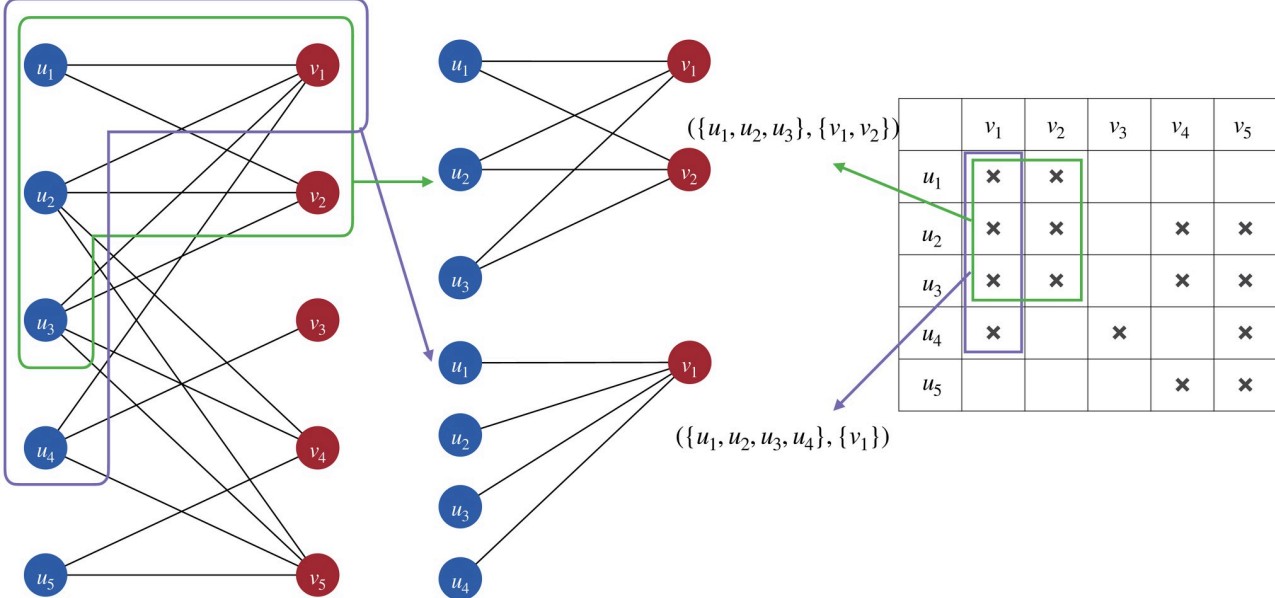

**Fig 3. A depiction of the equivalence between bipartite networks and formal contexts, as well as the equivalence between maximal bi-cliques and formal concepts.** The bipartite network to the left can be represented as the formal context to the right. The sub-network circled in purple and green are maximal bi-cliques in the bipartite networks to the left, which can be represented into two formal concepts framed in the corresponding colors in the formal context to the right.

## FCA and bipartite networks

By comparing the definitions above, we can easily find that the definition of a formal context is equivalent to that of a bipartite network, and the definition of a maximal bi-clique is equivalent to that of a formal concept. That is, for every bipartite network $C = (U, V, E)$, if we consider the two node sets of a bipartite network $U$ and $V$ as the object set and the attribute set, and the edge set $E$ as the binary relation of the objects and attributes, we may easily find that $(U, V, E)$ should also be a formal context. Also, if $C_1 = (A_1, B_1, E_1)$ is found to be a maximal bi-clique of $C$, it is certain that $(A_1, B_1)$ is also a formal concept in $\underline{\mathfrak{B}}(U, V, E)$. Fig 3 gives an example of such an equivalence between bipartite networks and formal contexts, as well as the equivalence between maximal bi-cliques and formal concepts.

Many previous research has spotted and discussed the two types of equivalences. However, besides these two types of equivalences, we may also find that the order relation and neighboring relation between formal concepts also correspond to the cover relations between maximal bi-cliques. The hierarchical order and neighboring relation between formal concepts are considered important in FCA and are represented by concept lattices, so in bipartite networks, the cover relations between maximal bi-cliques may also offer valuable insights for understanding the networks' structures. However, to the furthest of our knowledge, no previous studies on bipartite network theory have discussed on it. Hence, we believe that if we can design a model for learning the cover relations between bi-cliques, it will hopefully capture more information on the networks' structure and thus have a better performance in bipartite link prediction.

## BERT

*BERT*, short for *Bidirectional Encoder Representations from Transformers* [30], is a method for training a language model via the *Transformer* architecture [25]. It works in a "*pre-train* first

and *fine-tune* next" mode. First, it pre-trains a large language model with two general tasks on a large amount of unlabeled free text. Then, it fine-tunes the pre-trained model with a specific downstream task on a small labeled dataset.

The two general tasks used in the pre-training phase are *mask language model* (MLM) and *next sentence prediction* (NSP), which are defined as follows.

**MLM**: MLM is the task that predicts the full sentence from a sentence where some words are randomly masked with a special token "[MASK]". For example, if we have a sentence like "the quick brown fox jumps over the lazy dog", the model will take a masked version of the sentence like "the [MASK] brown fox jumps [MASK] the lazy dog" as input and the original sentence as target for output. The task helps the model learn the co-occurrence relationship between words in the same sentence [30].

**NSP**: NSP is the task that predicts whether two sentences are the subsequent sentences or not. The model takes a sentence pair (*A*, *B*) as input and is expected to output TRUE if *B* is the sentence directly following *A* in the corpus, or FALSE otherwise. The task helps the model to understand the relationships between sentences.

After pre-training, the values of the weight matrices of the pre-trained model are used as initial values for that of the fine-tune model and updated in the fine-tuning phase [31]. This is possible because pre-training and fine-tuning should use exactly the same network architecture except for the output layer. The downstream task in the fine-tuning phase should be the final target task. Hence, after fine-tuning, we get the final model which can be directly used for our target task.

## BERT and FCA

As mentioned above, BERT is a method for training language models, which takes sentences in natural languages as inputs. Although formal concepts are not sentences in natural languages, they do share some similarities—If we regard objects and attributes as words, then the extents and intents of a concept can be regarded as sentences in a language that has an order-free syntax. With such similarities, we may expect that the pre-training phase of BERT, which was originally designed for learning the features of words and sentences, can also be used for learning the features of objects, attributes, and formal concepts from a formal context and its corresponding concept lattice. We may also expect that bipartite link prediction can be a suitable downstream task after such a pre-training because the information of nodes (*a.k.a.* objects and attributes), and maximal bi-cliques (*a.k.a.* formal concepts) learned in the pre-training phase is helpful in increasing the accuracy of predictions.

Note that in most natural languages, the syntax is not order-free, indicating that different word orderings usually have completely different meanings. Hence, BERT has a special mechanism called *position embedding* for capturing the order of words in the input sentence. However, since the extents and intents of formal concepts are unordered, there is no need to keep track of the order. Hence, in our method, this mechanism is removed from BERT.

## Problem formulation and related work

### Problem formulation

The research deals with the problem of bipartite link prediction, which consists of two different tasks—the first task predicts the missing or unknown relation between two nodes from the same node set of a bipartite network, and the other task predicts the missing or unknown relation between two nodes from different node sets. As analyzed above, a bipartite network is equivalent to a formal context, and the two node sets are equivalent to the object set and the attribute set. Hence, for ease of understanding, we hereby name the two tasks as the *object-*

*object task* or the *O-O task* and the *object-attribute task* or the *O-A task*. The two node sets of a bipartite network are also directly called the *objects* and *attributes* in the rest part of the paper.

The formal definitions of the two tasks are defined as follows.

**O-O task**: Given an original network $C = (U, V, E)$ and a target network $C' = (U, V', E')$, the O-O task aims to predict if a group of nodes $G = \{u_1, u_2, \cdots, u_{|G|}\} \subseteq U$ that does not have *object-object links* in $C$, should have object-object links in $C'$. For a group of nodes $G \subseteq U$, they are considered to have *object-object links* in a network $(U, V, E)$ if $\exists v \in V$ such that $\forall u \in G$, $(u, v) \in E$.

**O-A task**: Given an original network $C = (U, V, E)$ and a target network $C' = (U, V, E')$, the O-A task aims to predict if two nodes $u \in U$ and $v \in V$ that does not have an *object-attribute link* in $C$, should have an object-attribute link in $C'$. For two nodes $u \in U$ and $v \in V$, they are considered to have an *object-attribute link* in a network $(U, V, E)$ if $(u, v) \in E$.

Note that the O-O task can be regarded as a special case of another task *hyperedge prediction*. For more details, please refer to related publications [32, 33].

## Related work on FCA-based bipartite link prediction

Previous methods for FCA-based bipartite link prediction can be classified into two groups—rule-based methods and embedding-based methods. The rule-based methods directly use pre-determined rules to make predictions based on the information provided by FCA; the embedding-based methods embed the objects and attributes into vectors using the information provided by FCA and use the embedding vectors for making link predictions.

In [17], the authors proposed a rule-based method for the O-A task by analyzing the overlapping formal concepts from a formal context. If the ratio of the overlapped part exceeds a predefined threshold, they are considered parts of the same large formal concept, so all the missing links in the non-overlapped part *a.k.a. the structure hole* [34] are predicted as present links. It presents an interesting approach to bipartite link prediction but only has a limited performance. In [19], the authors also used this approach, but only as a pre-processing step before the main link prediction step using matrix factorization. This hybrid method has much better results than the pure rule-based method.

In [7], the authors proposed an embedding-based method called *object2vec* to embed objects into vectors using the information from the formal concepts. Dually, they have also proposed *attribute2vec* for embedding attributes into vectors, which uses exactly the same mechanism as object2vec. It has two embedding models, *object2vec-CBoW* and *object2vec-SG*, both are derived from *Word2Vec* [22, 23]. Object2vec-CBoW, based on the *continuous-bag-of-words* model from Word2Vec, predicts a target object using objects around it within the same extent; object2vec-SG, based on the *skip-gram* model from Word2Vec, uses an object to predict other objects in the same extent. They conducted experiments on the O-O task on an author-publication network and demonstrated good performance.

In [20], the authors proposed another embedding method called *Bag of Attributes (BoA)*. Their method trains a more complicated embedding model using *Bidirectional Long Short-Term Memory* (Bi-LSTM) [24] and *Variational Autoencoder* (VAE) [35] on formal contexts. They conducted experiments on the O-O task on the same datasets as object2vec, and the results are similar to those obtained with object2Vec.

Both rule-based methods and embedding-based methods have some limitations. The rule-based methods use pre-determined rules, which may only work on some specific types of networks; the embedding-based methods use the information provided by FCA to train embedding vectors, but their model did not make full use of all the information provided by FCA—

for example, the ordering relation and neighboring relation of formal concepts are not learned into the embeddings.

## The proposed method

To address the limitations of the previous FCA-based methods for bipartite link prediction, we propose a novel method named *BERT4FCA*, which is designed to better learn the information of concept lattices and can conduct both the O-O task and the O-A task. We name our method *BERT4FCA* because it provides a general framework for using BERT to learn and utilize the information of concept lattices, so it is expected to be generally applicable to all tasks related to the context. Although in this research, we only discuss its application in the two bipartite link prediction tasks, we plan to study the possibility of applying our method to other tasks in the future.

Similar to all previous FCA-based bipartite link prediction methods, the objective of our method is to learn information from concept lattices and use it to make two link prediction tasks. However, the information our method aims to learn not only includes the extent and intent of a formal concept, but also includes the neighboring relations between formal concepts. Note that here we choose to learn the *neighboring relations* instead of the general *order relations* because, to reconstruct the concept lattice, the neighboring relations are enough [26].

Our method consists of 4 steps: *data preparation with FCA*, *input tokenization*, *BERT pretraining*, and *BERT fine-tuning*. An overview of the workflow of our method is shown in Fig 4.

### Data preparation

In this step, we convert the bipartite network into a formal context, extract all formal concepts, and construct the concept lattice. Then, we extract the neighboring relations between concepts

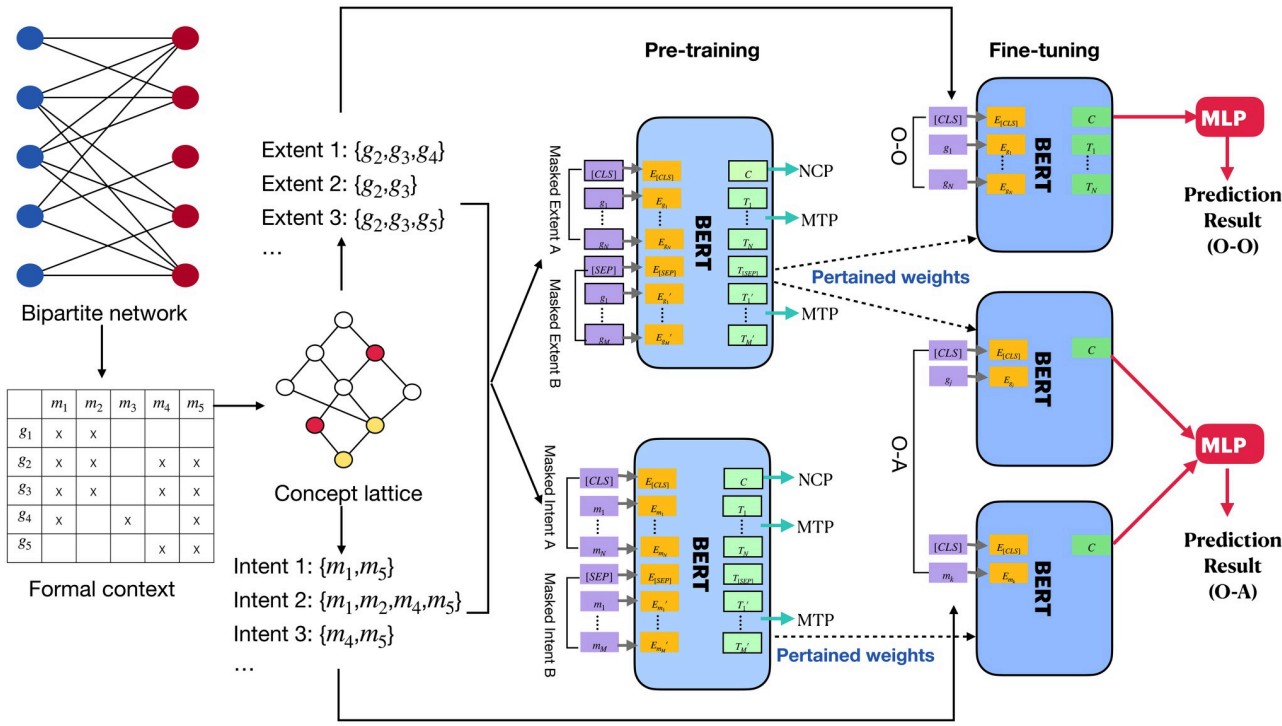

**Fig 4. An overview of the working flow of our method.**

from the concept lattice. We use Z-TCA [36] for extracting all formal concepts from a formal context, and use a *topological sorting* algorithm to extract neighboring relations from the concept lattice. The details of the algorithm are presented in Algorithm 1. After this step, we will obtain all extents and intents and neighboring concepts in the concept lattice.

**Algorithm 1** Get all neighboring relations using topological sorting.

```
Input A formal context 𝕂 = (U, V, E) and its concept lattice 𝕭(𝕂).
Output {N(C)}, a list of the lower neighbors of concepts. Here N(C)
represents the lower neighbor of C ∈ (𝕭(𝕂) − (∅, ∅″)).
1: for C ∈ 𝕭(𝕂) do
2:    D(C) ← |{C₁ ∈ 𝕭(𝕂) | C₁ < C}|
3: end for
4: Create Q as an empty queue.
5: for C ∈ 𝕭(𝕂) do
6:   if D(C) = 0 then
7:      Push C into the back of Q.
8:   end if
9: end for
10: while Q is not empty do
11:   Fetch C from the front of Q and pop it.
12:   for C₁ ∈ 𝕭(𝕂) such that C < C₁ do
13:      D(C₁) ← D(C₁) - 1
14:      if D(C₁) = 0 then
15:        N(C₁) ← C
16:        Push C₁ into the back of Q.
17:      end if
18:   end for
19: end while
```

## Input tokenization

In this step, the objects and attributes are tokenized into one-hot vectors. These one-hot vectors are further converted into dense vectors through the *input embedding* so that they can be processed by BERT.

The input embedding is the sum of two parts: *tokenization embedding* and *segment embedding*. Tokenization embedding is the general-sense embedding that uses a full-connect layer to embed the tokenized one-hot vectors into a dense vector space. Segment embedding is used to embed the tokens in the segment-info sequence into the vector space of the same dimension as the tokenization embedding. In BERT, sometimes the training sample may be a concatenation of two different sequences (details will be introduced later). In this case, the segment-info sequence is generated to distinguish the two different sequences within the same training sample. For example, in a training sample, the input is a sequence of seven tokens, where the first four tokens belong to the first sequence, and the last three tokens belong to the second sequence. Then, the segment-info sequence of this training sample should be (0, 0, 0, 0, 1, 1, 1).

Note that, as mentioned above, in the original framework of BERT, there is another input embedding called position embedding used for learning the order of words in a sentence. In BERT4FCA, however, it is removed because the input sequences are all unordered.

## BERT pre-training

In this step, we use the BERT framework to pre-train two models—the *object model* and the *attribute model*—on all extents and intents, correspondingly. Here, we only introduce the pre-training process of the object model for example. The input of a training sample of the pre-training of the object model is a pair of sequences, each representing an extent. Since the

lengths of extents may be different, we pad short extents to make all extents have the same length with a special token "[PAD]". We also add a special token, "[SEP]", between two sequences in order to separate them.

The model is trained with two tasks: *masked token prediction* (MTP) and *neighboring concepts prediction* (NCP), derived from the MLM and NSP tasks in the original version of BERT, correspondingly.

MTP is the task that helps the model learn the co-occurrence relationships between objects within the same extent. In the task, we randomly select a certain percentage of objects in both extents to be masked. For each object to be masked, we replace them into a special token "[MASK]" with 80% probability, or replace it with a random object with 10% probability, or keep it unchanged with 10% probability. Then, the model takes the masked pair of extents as input and the unmasked pair of extents as the target for output. That is, it is trained to predict the masked objects in the extents.

NCP is the task that helps the model learn the neighboring relations between formal concepts in the concept lattice. In the task, the model takes a pair of extents as input and is expected to output TRUE if the pair of extents corresponds to a pair of formal concepts that have neighboring relations or output FALSE otherwise. Note that in this task, clearly, the number of negative samples is much larger than the positive samples. To get a balanced training set, we randomly select a small portion of negative samples and keep the number of positive and negative samples to be the same.

The two tasks are trained simultaneously—the training loss is the sum of the losses of the two tasks. After pre-training, the pre-trained models are expected to have captured and stored information on the concept lattice, including relationships between objects and formal concepts.

**BERT Fine-tuning**: In this step, we fine-tune the pre-trained object model and attribute model to make them fit our target bipartite link prediction tasks.

For the O-O task, we fine-tune the pre-trained object model with the training samples generated from the original bipartite network ($U$, $V$, $E$). The training samples are generated with the following steps. First, we determine $l_m$, the maximal length of a group of objects we want to predict. Then, we enumerate all object subsets $U_1 \subseteq U$ such that $|U_1| \leq l_m$. For each $U_1$, we create a training sample that takes the tokenized and padded sequence of $U_1$ as input, and the label for the sample is set to TRUE if the objects in $U_1$ have object-object links and FALSE otherwise. After generation, the training samples are fed into a network that has the same structure as the network used in the pre-training step, except that in the last layer, the hidden states are not fed into the output layer used for the two pre-training tasks, but are fed into an output layer specified to the O-O task. Suppose the final output of the fine-tuning network for the O-O task is $P_{\text{O}-\text{O}}$; the last hidden state of the basic BERT network, *i.e.*, the "[CLS]" representation [30] is $h_{\text{L}}^{[\text{CLS}]}$, then the O-O-task-specified output layer can be described as follows:

$$P_{\text{O}-\text{O}} = \sigma(\text{ReLU}(h_{\text{L}}^{[\text{CLS}]} W_{\text{CLS}}) W), \tag{1}$$

where $\sigma(\cdot)$ is the sigmoid function; ReLU$(\cdot)$ is the *Rectified Linear Unit* (ReLU) [37]; $\mathbf{W}_{[\text{CLS}]}$ and $\mathbf{W}$ are weight matrices.

For the O-A task, we fine-tune both the pre-trained object model and the attribute model together with training samples generated from the original bipartite network. The training samples are generated with the following steps. We first enumerate all objects $u \in U$ and attributes $v \in V$. For each pair of $(u, v)$, we create a training sample that takes the tokens of $u$ and $v$ as input, and the label for the sample is set to TRUE if $(u, v) \in E$ and FALSE otherwise. After generation, the tokenized object and tokenized attribute are fed into two separate BERT

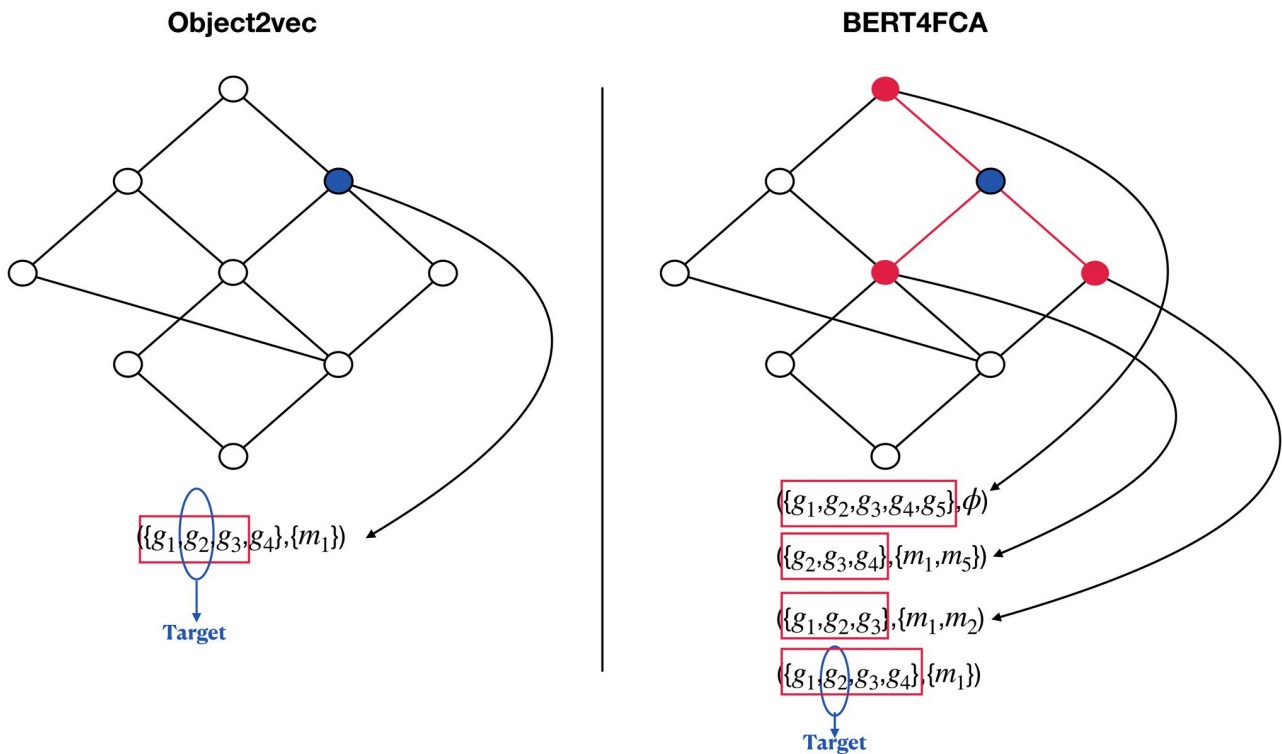

**Fig 5. The comparison of how much information from a concept lattice is learned and used when predicting an object by two methods, with object2vec shown on the left and BERT4FCA shown on the right.** The target object to be predicted is circled in blue. The information used for predicting the object is shown in red.

networks—except for the last output layer, the first network has the same structure as the network used in pre-training the object model, and the second network has the same structure as that used in pre-training the attribute model. The last hidden states of both networks are concatenated and fed through a single output layer specified for the O-A task. Suppose the final output of the fine-tuning network for the O-A task is $P_{\text{O}-\text{A}}$; the last hidden states of the two BERT networks are $\boldsymbol{h}_{\text{L1}}^{\text{[CLS]}}$ and $\boldsymbol{h}_{\text{L2}}^{\text{[CLS]}}$, then the O-A-task-specified output layer can be described as follows:

$$P_{\text{O}-\text{A}} = \sigma\big(\text{ReLU}\big((\boldsymbol{h}_{\text{L1}}^{\text{[CLS]}} {}^\frown \boldsymbol{h}_{\text{L2}}^{\text{[CLS]}})\boldsymbol{W}_{\text{CLS}}\big)\boldsymbol{W}\big), \tag{2}$$

where $\mathbf{a}^\frown\mathbf{b}$ means the concatenation of vectors $\mathbf{a}$ and $\mathbf{b}$.

Above is the whole workflow of our method. From above, we can see that our method has three advantages. First, in the pre-training step, our method can learn more information from concept lattices compared to the previous FCA-based embedding methods like object2vec. As shown in Fig 5, in object2vec, when embedding an object, it uses only the information of a small set of objects within the same extent. In the pre-training step of our method, however, we learn the feature of an object using the information from all objects in the entire extent as well as the objects in the extents of its neighbor concepts in concept lattice.

Second, our method can directly extract useful features for link prediction from the formal context, in addition to learning from the concept lattice. During fine-tuning step, we train the model with samples generated from the formal context, which enables the model to learn directly from formal contexts. Therefore, even in datasets where the concept lattice cannot

provide much useful information for link prediction, our method can still learn from the formal contexts for link prediction. In contrast, previous FCA-based methods do not possess the capability of directly learning from the formal context. Additionally, previous non-FCA-based methods for bipartite link prediction cannot make use of bi-cliques for link prediction.

Third, our method works in a pre-train first and fine-tune next mode that first pre-trains two large models and then fine-tunes them on various downstream tasks related to the formal context, meaning that if we wish to conduct new tasks on a formal context where we have pre-trained the object model and attribute model, we can skip the first three steps and directly conduct the final fine-tuning step. Previous methods, however, may need to re-train the models when coming to a new task.

## Experiments

### Datasets

We conduct experiments on five real-world datasets: *ICFCA*, *BMS-POS*, *Keyword-Paper*, *Review* and *iJO1366*. All datasets are used for both the O-O task and the O-A task. We depict the features of these datasets in Table 1. A detailed description of each follows.

**ICFCA.** The ICFCA dataset is an author-paper network provided by [7]—the objects represent the authors, the attributes represent the publications, and each edge (*a.k.a.* relation) represents the author is in the author list of the publication. This dataset is generated from *Digital Bibliography & Library Project* (DBLP) dump on 1st Aug 2019 which is available at https://dblp.uni-trier.de/xml/.

For the O-O task, we are to simulate the practical case in which we wish to predict future co-authorships or seek potential co-authors from an author-paper network at a certain time point. Hence, we generate a history network from the full network as the input network for

**Table 1. The features of the datasets.**

| Dataset | Task | Input/Target | Objects | Attributes | Edges | Concepts |
|---|---|---|---|---|---|---|
| ICFCA | O-O | Input | 334 | 12614 | 13399 | 775 |
| | | Target | 334 | 12614 | 15980 | 844 |
| | O-A | Input | 351 | 12614 | 14445 | 878 |
| | | Target | 351 | 12614 | 16049 | 922 |
| BMS-POS | O-O | Input | 468 | 1946 | 7376 | 7791 |
| | | Target | 468 | 1946 | 8085 | 10235 |
| | O-A | Input | 468 | 1946 | 7376 | 7791 |
| | | Target | 468 | 1946 | 8085 | 10235 |
| Keyword-Paper | O-O | Input | 162 | 5640 | 7274 | 1610 |
| | | Target | 162 | 5640 | 8308 | 2049 |
| | O-A | Input | 162 | 5206 | 7648 | 1713 |
| | | Target | 162 | 5206 | 7907 | 2046 |
| Review | O-O | Input | 181 | 340 | 420 | 262 |
| | | Target | 181 | 340 | 465 | 281 |
| | O-A | Input | 181 | 340 | 420 | 262 |
| | | Target | 181 | 340 | 465 | 281 |
| iJO1366 | O-O | Input | 1805 | 2583 | 9231 | 5185 |
| | | Target | 1805 | 2583 | 10184 | 5595 |
| | O-A | Input | 1805 | 2583 | 9231 | 5185 |
| | | Target | 1805 | 2583 | 10184 | 5595 |

this task by removing the authors, publications, and author-paper edges after 1st Jan 2016; we generate the current network as the target network for this task from the full network by removing authors who had no publication before 31st Dec 2015 and their corresponding edges.

For the O-A task, we are to simulate the practical case that some parts of the network are missing, and we wish to use the known edges in the network to predict the potentially missing edges. We generate the input network for this task from the full network by randomly removing 10% of author-paper edges; we use the full network as the target network for this task.

**BMS-POS.** The BMS-POS dataset is a product purchased transactions network provided by KDD Cup 2000—the objects represent the products, the attributes represent purchasing transactions, and an edge represents that the product is bought in a certain purchasing transaction. The original data is very large, so in this research, we only select the first 1946 transactions. The dataset is available at https://kdd.org/kdd-cup/view/kdd-cup-2000.

For the O-O task, we are to simulate the practical case in which we wish to predict two products will be likely to be bought by the same customer. Hence, we generate the input network from the full network by randomly removing 10% of product-transaction edges; we use the full network as the target network for this task.

For the O-A task, we are to simulate the same practical case as the ICFCA dataset. We use the same input network and target network as those used in the O-O task of this dataset.

**Keyword-paper.** The Keyword-Paper dataset is an original dataset generated by us—the objects represent the keywords, the attributes represent the publications, and each edge represents the paper has the keyword. It is generated from the DBLP dump on 31st Jan 2023. From the dump, we select the top 162 most frequent keywords and all 5640 publications after 1st Jan 2010 to create the keyword-paper network.

For the O-O task, we are to simulate the practical case in which we wish to predict potentially related keywords, which may give inspiration for new research. For example, while "BERT" and "FCA" were never used as keywords of the same paper before, if they are predicted to be related, researchers may get inspired and draft a new study similar to ours. The generation of the input network and target network are similar to that of the ICFCA dataset, and the date for the history network is also set to 31st Dec 2015.

For the O-A task, we are to simulate the same practical case as the previous two datasets. We also use the same way to generate the input and target networks as the way we used in the O-A tasks of the previous two datasets. Note that in this dataset, after removing 10% of edges, some attributes will have no edge connecting to them, so the number of attributes in the networks used for the O-A task is smaller than that used for the O-O task.

**Review.** The Review dataset is a peer review data from a top computer science conference. It is collected from [38] and previously used as a test dataset in [39]. The objects represent reviewers; the attributes represent manuscripts; and each edge represent that the reviewer suggests accepting the manuscript.

For both the O-O and the O-A tasks, we are to simulate the practical cases similar to those of the BMS-POS dataset. Hence, the input networks and the target networks are also generated using the same procedures as for the BMS-POS dataset.

**iJO1366.** The iJO1366 dataset is a metabolite-reaction network processed from a genome scale metabolic network of the Escherichia coli. It is collected from the BiGG dataset [40] and previously used as a test dataset in [41]. We extract the metabolites as objects, and the reactions as attributes, and each edge represents that the metabolite participates in the reaction.

For both the O-O and the O-A tasks, we are to simulate the practical cases similar to those of the BMS-POS dataset. Hence, the input networks and the target networks are also generated using the same procedures as for the BMS-POS dataset.

## Generation of labeled training and test samples

For each dataset, we have generated an input network $C = (U, V, E)$ and the target network $C'$ = $(U, V, E')$. We are then to generate labeled training and test samples from the two networks with the following procedure.

**The O-O task.** For the training samples, we first enumerate all object-object links in $C$ which contains no more than $\varepsilon_p$ objects, with $\varepsilon_p$ being a pre-determined threshold. These links are labeled as positive training samples. That is, we are to enumerate every object group $G \subseteq U$ satisfying that $\|G\| \leq \varepsilon_p$ and $\exists v \in V$ such that $\forall u \in G, (u, v) \in E$. Then, we are to randomly choose the same number of negative training samples samples as the positive training samples, with each negative sample being an object-object link which has no more than $\varepsilon_p$ objects and does exists in $C$. Formally, each negative sample should be an object group $G \subseteq U$ such that $\|G\| \leq \varepsilon_p$ and $\nexists v \in V$ such that $\forall u \in G, (u, v) \in E$. The proportion of object-object links with different sizes in the set of negative training samples is kept the same as that in the set of positive training samples. That is, suppose the set of all positive training samples is $T_1$, and the set of all negative training samples is $T_2$, we should have $\|T_1\| = \|T_2\|$ and for each $i = 2, 3, \cdots, \varepsilon_p$, we should have $\|\{G \in T_1 \| G\| = i\}\| = \|\{G' \in T_2 \| G'\| = i\}\|$.

For the test samples, we enumerate all object-object links in $C'$ which contains no more than $\varepsilon_p$ nodes and **does not appear in $C$**. These links are labeled as positive test samples. That is, we are to extract every object group $G \subseteq U$ satisfying that $\|G\| \leq \varepsilon_p$ and $\exists v \in V$ such that $\forall u \in G, (u, v) \in E'-E$. Then, we are to randomly choose the same number of negative test samples samples as the positive test samples, with each negative test sample being an object-object link which has no more than $\varepsilon_p$ objects and does exists in $C'$. Formally, each negative test sample is a group $G \subseteq U$ such that $\|G| \leq \varepsilon_p$ and $\nexists v \in V$ such that $\forall u \in G, (u, v) \in E'$. The proportion of object-object links with different sizes in the set of negative test samples is also kept the same as that in the set of positive test samples.

In this research, the value of $\varepsilon_p$ is set to 5 for the ICFCA and Review datasets; it is set to 3 for the other three datasets. Although theoretically, our method as well as all previous methods can predict object-object links containing any number of objects, we set this limit to prevent the number of training/test samples from exponentially growing.

**The O-A task.** For the training samples, we first enumerate all object-attribute links in $C$ and label them as positive training samples. That is, we are to enumerate every object-attribute pair $(u, v)$ such that $u \in U, v \in V$ and $(u, v) \in E$. Then, we are to randomly generate the same number of negative training samples as the positive training samples, with each negative training sample being an object-attribute link which does not exist in $C$. Formally, each negative training sample $(u, v)$ should satisfy $u \in U, v \in V$ and $(u, v) \notin E$.

For the test samples, we first enumerate all object-attribute links in $C'$ which does not exist in $C$. These links are all labeled as positive test samples. That is, we are to enumerate every pair $(u, v)$ such that $u \in U, v \in V$ and $(u, v) \in E' - E$. Then, we are to randomly generate the same number of negative training samples as the positive training samples, with each negative training sample being an object-attribute link which does not exist in $C'$. Formally, each negative test sample $(u, v)$ should satisfy $u \in U, v \in V$ and $(u, v) \notin E'$.

## Evaluation metrics

To give a fair and comprehensive evaluation, we use the following three measures: $F_1$ *score*, *AUC score*, and *AUPR score*. All three scores are estimated with the four basic values: TP, TN, FP, and FN. TP represents the number of samples that are positive and are predicted positive. FN represents the number of samples that are negative but are falsely predicted to be positive.

TN represents the number of samples that are negative but are predicted negative. FN represents the number of samples that are positive but falsely predicted negative.

The $F_1$ score is the harmonic mean of the *precision* and *recall*. Precision, recall, and $F_1$ are estimated as follows:

$$
\begin{aligned}
Percision &:= \frac{TP}{TP + FP}, \\
Recall &:= \frac{TP}{TP + FN}, \\
F_1 &:= \frac{2}{recall^{-1} + percision^{-1}}.
\end{aligned}
\tag{3}
$$

The $F_1$ value may vary as we change the threshold for the prediction score. Hence, in this research, for each test case, we try 20 different thresholds with the following procedure—the initial threshold is set to 0, and at each trial, we add the threshold by 0.05. After all trials, we report the highest $F_1$ value we get.

The AUC (Area Under the Curve) score is estimated by computing the area under the ROC (Receiver Operating Characteristic) curve. The ROC curve is created by plotting the true positive rate (TPR) against the false positive rate (FPR) at various threshold settings. TPR and FPR are estimated as follows:

$$
\begin{aligned}
TPR &:= \frac{TP}{TP + FN}, \\
FPR &:= \frac{FP}{TN + FP}.
\end{aligned}
\tag{4}
$$

The AUPR (Area Under the Precision-Recall Curve) score is estimated by computing the area under the Precision-Recall curve, which is created by plotting the precision rate against the *recall* rate at various threshold settings.

## Experimental environment and parameters

**Experimental environment.** The experiments are conducted on a Windows 11 server with 64GB RAM, an AMD Ryzen 9 7900X CPU, and an NVIDIA GeForce RTX 4090 GPU.

The codes for the deep-learning part of our method are implemented with Python 3.8.18 and Pytorch 2.1.0. The code for the algorithm for extracting formal concepts is written in C++ and complied by MinGW 13.2.0.

**Parameters.** The dimension of input embeddings of the Transformer encoder is 768, the dimension of hidden layer in Transformer encoder is 3072.

For the BERT pre-training step, the maximum number of masked tokens used in MTP is set to 4, the dimension of embeddings is set to 768, the number of heads is set to 12, the number of Transformer encoders is set to 6.

For the BERT fine-tuning step, the dimension of the hidden layer is set to 512. The batch size is set to 24.

## The experiment of the O-O task

We evaluate the performance of BERT4FCA in the O-O task on the five datasets. The competitor methods used in this experiment include an FCA-based method *object2vec* [7], a widely-used classic method *Node2Vec* [13], two widely-used classic hyperedge prediction methods,

**Table 2. The results for the O-O tasks.**

| Method | ICFCA | | | BMS-POS | | | Keyword-Paper | | | Review | | | iJO1366 | | |
|---|---|---|---|---|---|---|---|---|---|---|---|---|---|---|---|
| | $F_1$ | AUC | AUPR | $F_1$ | AUC | AUPR | $F_1$ | AUC | AUPR | $F_1$ | AUC | AUPR | $F_1$ | AUC | AUPR |
| O2V-CBoW | 0.686 | 0.691 | 0.672 | 0.690 | 0.637 | 0.560 | 0.463 | 0.504 | 0.518 | 0.614 | 0.543 | 0.549 | 0.579 | 0.581 | 0.527 |
| O2V-SG | 0.652 | 0.497 | 0.519 | 0.676 | 0.649 | 0.577 | 0.236 | 0.290 | 0.388 | 0.527 | 0.501 | 0.480 | 0.612 | 0.674 | 0.675 |
| Node2Vec | 0.587 | 0.703 | 0.751 | 0.870 | 0.946 | 0.945 | 0.438 | 0.647 | 0.635 | 0.667 | 0.611 | 0.613 | 0.671 | 0.630 | 0.621 |
| HPRA | 0.752 | 0.808 | 0.767 | 0.862 | 0.927 | 0.824 | 0.722 | 0.761 | 0.675 | 0.675 | 0.724 | 0.668 | 0.956 | 0.986 | 0.889 |
| CN | 0.741 | 0.794 | 0.777 | 0.872 | 0.929 | 0.824 | 0.729 | 0.784 | 0.701 | 0.667 | 0.709 | 0.684 | 0.962 | 0.986 | 0.889 |
| Katz | 0.752 | 0.779 | 0.754 | 0.853 | 0.918 | 0.814 | 0.724 | 0.773 | 0.688 | 0.686 | 0.726 | 0.681 | 0.819 | 0.748 | 0.757 |
| BERT4FCA | **0.781** | **0.877** | **0.896** | **0.871** | **0.964** | **0.963** | **0.800** | **0.911** | **0.908** | **0.765** | **0.788** | **0.758** | **0.965** | **0.994** | **0.994** |

Note that O2V stands for object2vec. CN stands for common neighbors.

*common neighbors* [42] and *Katz index* [43], and a relatively new hyperedge prediction method *HPRA* [32]. Note that for the object2vec method, both CBoW and SG models are tested. For the Node2Vec method, after obtaining embedding vectors for each object and attribute from the input network, we train a downstream Logistic Regression classifier using the labeled training samples generated with the procedure introduced above. For each sample $G$, its feature is set to the concatenation of the embedding vectors of every object $g \in G$. The objects are sorted in the lexicographical order before concatenation. If a sample has fewer than $\varepsilon_p$ objects, its feature vector is padded with zeros to make sure that all samples' feature vectors are of the same dimension. For the three hyperedge prediction methods, as introduced previously, they can be directly applied to the O-O task simply because the O-O task is a special case of hyperedge prediction.

The results are shown in Table 2.

The results show that BERT4FCA outperforms the other models across all datasets. We have also discovered that although Node2Vec has achieved high scores close to our method on the BMS-POS dataset, its performances on the other datasets are much lower than our method. This shows the stability of our method, *i.e.*, we can have a stable and high performance across different datasets. Also, as can be seen from the result, the three hyperedge prediction methods' performance are far better than object2vec, while our BERT4FCA can still outperform all of them. This implies that the ideology of utilizing the information in concept lattice for link prediction is valuable, while object2vec does not utilizes it very well. Our BERT4FCA, on the other hand, has fully learned and utilizes the information from the concept lattice and thus has achieved good results. We will conduct extra ablation experiments later in this section to prove that our inference is correct.

## The experiment of the O-A task

We evaluate the performance of BERT4FCA in the O-A task on all datasets. We compare BET4FCA with an FCA-based method, *Structure Hole* [17], two widely-used non-FCA-based classic methods—*Node2Vec* and *matrix factorization with singular value decomposition* (MF-SVD), and a relatively new deep-learning method, *SBGNN* [39]. For the Node2Vec method, after obtaining embedding vectors for each object and attribute from the input network, we train a downstream Logistic Regression classifier using the labeled training samples generated with the procedure introduced above. For each sample $(u, v)$, its feature is set to the concatenation of the embedding vectors of object $u$ and attribute $v$.

The results are shown in Table 3.

**Table 3. The results for the O-A tasks.**

| Method | ICFCA | | | BMS-POS | | | Keyword-Paper | | | Review | | | iJO1366 | | |
|---|---|---|---|---|---|---|---|---|---|---|---|---|---|---|---|
| | $F_1$ | AUC | AUPR | $F_1$ | AUC | AUPR | $F_1$ | AUC | AUPR | $F_1$ | AUC | AUPR | $F_1$ | AUC | AUPR |
| Structure Hole | 0.018 | 0.000 | 0.000 | 0.226 | 0.000 | 0.000 | 0.005 | 0.000 | 0.000 | 0.000 | 0.000 | 0.000 | 0.000 | 0.000 | 0.000 |
| Node2Vec | 0.671 | 0.800 | 0.764 | 0.612 | 0.793 | 0.755 | 0.661 | 0.747 | 0.697 | 0.630 | 0.579 | 0.566 | 0.690 | **0.757** | **0.820** |
| MF-SVD | **0.749** | 0.798 | 0.508 | **0.824** | **0.892** | 0.641 | 0.733 | 0.693 | 0.424 | 0.531 | 0.487 | 0.069 | 0.630 | 0.730 | 0.549 |
| SBGNN | 0.701 | 0.710 | 0.652 | 0.816 | 0.803 | 0.731 | 0.658 | 0.655 | 0.615 | 0.623 | 0.556 | 0.530 | 0.602 | 0.700 | 0.679 |
| BERT4FCA | 0.741 | **0.812** | **0.777** | 0.725 | 0.823 | **0.788** | **0.744** | **0.765** | **0.711** | **0.677** | **0.675** | **0.690** | **0.697** | 0.741 | 0.757 |

From the results, we can see that among all five datasets, BERT4FCA has the best overall performance. Its performance is notably better than Structure Hole and generally better than SBGNN. For MF-SVD and Node2Vec, although they have achieved higher scores than BERT4FCA in some metrics in some dataset, their overall performance is mixed, while our BERT4FCA has an overall good performance acorss five datasets. The reason why these two methods get mixed performance is analyzed below.

In the BMS-POS dataset, MF-SVD exhibits higher AUC scores than other methods but with very low AUPR scores. A low AUPR score indicates that the model's high prediction scores do not correlate well with being in the positive class, suggesting that the model has difficulty achieving high precision. Since in link prediction, we focus on predicting the generation of new links, which is predicting the positive samples but not the negative samples, so if a model exhibits a low AUPR, it is considered to have low performance in link prediction. The low performance of MF-SVD may suggest that the underlying latent features of the networks and the relationships between nodes are intricate and challenging for MF to capture effectively. Additionally, since the BMS-POS dataset is extracted from a large dataset, maybe the relationships between nodes in this network are not consistent and stable, resulting in low prediction performance of MF.

In the iJO1366 dataset, Node2Vec exhibits higher AUPR and AUC scores than our method. However, its scores in the Review dataset is far lower than our method, indicating that the performance of Node2Vec varies greatly across different datasets. According to previous research [44], the sparser the network is, the better the process can capture the features of the network. Hence, it is expected to work better on sparse networks. Since iJO1366 is the most sparse data set, Node2Vec achieved to gain a better performance, but for relative dense networks, its performances become notably lower. Hence, we consider our method to be more practical than Node2Vec thanks to its overall good performance in all metrics across all datasets.

Finally, we have also found that the performance of the other FCA-based method, Structure Hole, is notably lower on these five datasets than the datasets used in [17]. As analyzed before, Structure Hole is a rule-based method that uses the same simple rule to extract information from the concept lattices and make predictions on all datasets, which will be highly likely to have lower performance on some datasets because not all datasets fit the rule well, and our experimental results just proved it. This again shows the importance of developing a method like our BERT4FCA that can automatically capture the information from concept lattices using statistical machine-learning techniques.

## Ablation experiments

To verify if our original mechanisms are functioning well, we conduct two ablation experiments—the first one checks if our method can indeed learn the neighboring relations of formal

**Table 4. The results for the first supplementary experiment.**

| Dataset | Object/Attribute | $F_1$ | AUC | AUPR |
|---|---|---|---|---|
| ICFCA | Object | 0.903 | 0.896 | 0.842 |
| | Attribute | 0.868 | 0.842 | 0.785 |
| BMS-POS | Object | 0.993 | 0.993 | 0.987 |
| | Attribute | 0.978 | 0.978 | 0.963 |
| Keyword-Paper | Object | 0.965 | 0.966 | 0.934 |
| | Attribute | 0.900 | 0.902 | 0.850 |
| Review | Object | 0.938 | 0.826 | 0.882 |
| | Attribute | 0.857 | 0.706 | 0.786 |
| iJO1366 | Object | 0.977 | 0.990 | 0.956 |
| | Attribute | 0.880 | 0.918 | 0.796 |

concepts; the second one checks if the information in concept lattices our method learned is indeed helpful for making link predictions.

In the first ablation experiment, we only use 80% of the training samples to pre-train both the object model and the attribute model on all datasets. The remaining 20% of the original training samples are kept as test samples for the NCP task—these intents/extents pairs are not presented in the training samples, so if our method can correctly predict whether they are neighbors or not, it should indicate that our method has learned the structure of concept lattice well. The results are shown in Table 4.

The results suggest that BERT4FCA indeed learned the neighboring relations from the concept lattice on all datasets well. We have also noticed that the results of the object models are generally better than the attribute models across all datasets. We analyze it because, in these datasets, the average lengths of intents are longer than that of the extents, making it potentially more challenging to effectively capture the neighboring relations of intents.

In the second ablation experiment, we check if the information we learned from concept lattices indeed contributes to better link prediction results. Since the information of the concept lattices is learned in the NCP and MTM tasks in the pre-training step, in this experiment, we skip the pre-training step and directly fine-tune the models with randomly initialized weights. By comparing the results with that of our full method, we will know whether our specially designed tasks for learning the information of concept lattices are functioning well. The results for the O-O tasks are shown in Table 5, and the results for the O-A tasks are shown in Table 6.

The results show that learning more information from concept lattices indeed contributes to the improvement of performances in both tasks of bipartite link prediction, while the degree of improvement varies across different datasets and tasks. The overall degree of improvement in O-O tasks is higher than that in the O-A task. We analyze it because compared to the attribute models, the object models better capture the information on concept lattices, such as the

**Table 5. The results for the O-O tasks of the second ablation experiment.**

| Method | ICFCA | | | BMS-POS | | | Keyword-Paper | | | Review | | | iJO1366 | | |
|---|---|---|---|---|---|---|---|---|---|---|---|---|---|---|---|
| | $F_1$ | AUC | AUPR | $F_1$ | AUC | AUPR | $F_1$ | AUC | AUPR | $F_1$ | AUC | AUPR | $F_1$ | AUC | AUPR |
| BERT4FCA-NC | 0.718 | 0.810 | 0.815 | 0.865 | 0.961 | 0.958 | 0.735 | 0.866 | 0.858 | 0.719 | 0.730 | 0.719 | 0.940 | 0.985 | 0.987 |
| BERT4FCA | **0.781** | **0.877** | **0.896** | **0.871** | **0.964** | **0.963** | **0.800** | **0.911** | **0.908** | **0.765** | **0.788** | **0.758** | **0.965** | **0.994** | **0.994** |
| Improvement | 0.063 | 0.067 | 0.081 | 0.006 | 0.003 | 0.005 | 0.065 | 0.045 | 0.050 | 0.046 | 0.058 | 0.039 | 0.025 | 0.009 | 0.007 |

**Table 6. The results for the O-A tasks of the second ablation experiment.**

| Method | ICFCA | | | BMS-POS | | | Keyword-Paper | | | Review | | | iJO1366 | | |
|---|---|---|---|---|---|---|---|---|---|---|---|---|---|---|---|
| | $F_1$ | AUC | AUPR | $F_1$ | AUC | AUPR | $F_1$ | AUC | AUPR | $F_1$ | AUC | AUPR | $F_1$ | AUC | AUPR |
| BERT4FCA-NC | 0.723 | 0.801 | 0.766 | 0.701 | 0.800 | 0.760 | 0.724 | 0.748 | 0.707 | 0.667 | 0.660 | 0.629 | 0.695 | 0.740 | 0.749 |
| BERT4FCA | **0.741** | **0.812** | **0.777** | **0.725** | **0.823** | **0.788** | **0.744** | **0.765** | **0.711** | **0.677** | **0.675** | **0.690** | **0.697** | **0.741** | **0.757** |
| Improvement | 0.018 | 0.011 | 0.011 | 0.024 | 0.023 | 0.028 | 0.020 | 0.017 | 0.004 | 0.010 | 0.015 | 0.061 | 0.002 | 0.001 | 0.008 |

neighboring relations, as demonstrated in the first ablation experiment. The difference in improvements shows that better learning the neighboring relations contributes to the prediction performance.

Also, we find that in the O-O task on the BMS-POS dataset, learning the information from concept lattices contributes to the least degree of improvement. This may be because the performance of our method without learning the information is already sufficiently high, as the AUC and AUPR scores have reached 96%. In such a case, it is considered hard to improve the performance further, even with an effective mechanism.

Furthermore, we observe that even without the pre-training step, our method can still have a relatively good performance. This indicates the BERT framework is capable of extracting information to make predictions even directly from the raw bipartite network. This discovery may provide hints for further research on the possibility of training a light-weight model with a small part of the network utilizing the effective BERT-like transformer network structure. Nevertheless, when there is an urge requirement of stability and quality, it is recommended to apply our full BERT4FCA model as the information from maximal bi-cliques extracted by our proposed MTP and NCP tasks can indeed further improve the performance.

## Conclusion and future work

In this paper, we proposed BERT4FCA, a novel FCA-based method for bipartite link prediction. It can learn more information from concept lattices and then use it to make bipartite link prediction. The experimental results demonstrated that our methods outperform previous FCA-based methods like object2Vec and non-FCA-based classic methods like MF and Node2-Vec in both the O-O and the O-A tasks on five different datasets. The results have shown that our method is stabler than previous methods. We also demonstrated by ablation experiments that neighboring relations between maximal bi-cliques are well learned by the model as expected and that such information contributes to better link prediction results. Furthermore, we have shown that BERT4FCA provides a general framework for employing BERT to learn the information extracted by FCA. Hence, in addition to bipartite link prediction, we believe BERT4FCA can also be further applied to a broader range of real-world tasks. We plan to explore the potential applications in the future.

However, we have also found some points of our method that may be further refined and improved. First, limited by computational resources, we were unable to pre-train an all-in-one model for both objects and attributes in this research. If such a model can be trained well, it will be more convenient as all downstream tasks will only need one single pre-trained model. Second, although BERT4FCA outperforms previous methods in prediction results, the pre-training step is time-consuming, especially when dealing with datasets with a large number of formal concepts. With our current computational resources, to finish the pre-training step in a reasonable time, the maximum number of concepts should be around 15,000. Although there have been methods for reducing the number of formal concepts in concept lattices, it is uncertain whether training on a reduced concept lattice will result in lower performances. If so, we

may need to work out a technique to make a trade-off between the training time and the performance. We plan to address these two points in our future work.

## Author Contributions

**Conceptualization:** Siqi Peng, Hongyuan Yang.

**Funding acquisition:** Siqi Peng, Akihiro Yamamoto.

**Methodology:** Siqi Peng, Hongyuan Yang.

**Supervision:** Akihiro Yamamoto.

**Validation:** Siqi Peng, Hongyuan Yang.

**Writing – original draft:** Siqi Peng, Hongyuan Yang.

**Writing – review & editing:** Akihiro Yamamoto.

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
