## [Decision Letter · Decision Letter 0]

26 Mar 2024

PONE-D-24-07133BERT4FCA: A Method for Bipartite Link Prediction using Formal Concept Analysis and BERTPLOS ONE

Dear Dr. Yang,

Thank you for submitting your manuscript to PLOS ONE. After careful consideration, we feel that it has merit but does not fully meet PLOS ONE’s publication criteria as it currently stands. Therefore, we invite you to submit a revised version of the manuscript that addresses the points raised during the review process.

We look forward to receiving your revised manuscript.

Kind regards,

Karthik Raman, Ph.D.

Academic Editor

PLOS ONE

Journal Requirements:

"This study was funded by the Japan Society for the Promotion of Science (https://www.jsps.go.jp/english/e-grants/)in the form of a Grant-in-Aid for Socientific Research(B) to SP and AY[JP21H03499]."

**Additional Editor Comments:**

The reviews for your manuscript are now in, and while the reviewers appreciate the work, they have raised several concerns that will need to be addressed in a revised version of the manuscript.

Reviewers' comments:

Reviewer's Responses to Questions

**Comments to the Author**

1. Is the manuscript technically sound, and do the data support the conclusions?

Reviewer #1: Yes

Reviewer #2: Yes

2. Has the statistical analysis been performed appropriately and rigorously? 

Reviewer #1: Yes

Reviewer #2: No

3. Have the authors made all data underlying the findings in their manuscript fully available?

Reviewer #1: Yes

Reviewer #2: Yes

4. Is the manuscript presented in an intelligible fashion and written in standard English?

Reviewer #1: Yes

Reviewer #2: Yes

5. Review Comments to the Author

Reviewer #1: This paper presents an interesting proposal, Bipartite Link Prediction. However, there is room for improvement before the manuscript can be accepted for publication. I recommend that the authors carefully revise the manuscript based on the following feedback:

1. I strongly suggest that the authors enhance the clarity of their Abstract by clearly stating their motivation for the proposed method.

2. While the experiments conducted are sufficient, further analysis could significantly enhance the contributions and claims made in the study.

3. It is incremental work; I still want to know the main difference and the new.

4、 In the experiment section, the latest baselines cited were published in 2020. The authors need to add some algorithms from the last three years as baselines.

5. How was the experiment implemented, with MATLAB or Python? Will the code be publicly available?

6.Provide more information about how the experiments were conducted, including the tools and software used, in the experiment section.

7. Provide diagram for proposed method.

8. the authors should analyse how to set the parameters of the proposed methods in the framework. Do they have the “optimal” choice?

9.The Literature citation is not adequate, and the related work should be discussed:

9.1 A novel link prediction algorithm for protein-protein interaction networks by attributed graph embedding

9.2. Autoencoders and their applications in machine learning: a survey

Reviewer #2: Summary: The authors introduce BERT4FCA, a method designed for predicting links in bi-partite networks. The proposed method works on the principles of equivalence between concepts from bi-partite networks (links and bi-cliques) and FCA (formal context and concept). Authors have shown that the proposed method outperforms the baselines considered.

Strengths:

1. BERT4FCA addresses the challenge of predicting links between existing node sets as well as new relations where some nodes have not been observed before. This flexibility is crucial and adds value to the literature, given the limited attention this problem has received previously.

2. Results indicate that BERT4FCA outperforms the baseline methods considered, demonstrating its effectiveness in link prediction tasks.

3. The paper is well-written and easy to follow, albeit minor typos.

Weaknesses:

1. Unclear novelty: The exact novelty of the proposal and what exactly derives superior performance of the proposed method is not clearly laid out.

2. Missing prominent baselines: Notably, several relevant baselines from related disciplines, such as hyperedge prediction in hypergraphs, have been overlooked. These baselines could provide valuable insights and should be included for a comprehensive evaluation (follow the references at end).

3. Limited Dataset and Ablation Study: The datasets considered are very few in numbers and very small in the size. Also, there is no ablation study which makes it hard to understand how the proposed method works and leads to these results.

Questions:

1. In the definition of Concept Lattices, shouldn't it be $B_1 \\subset \\B_2$?

2. Is the O-O task limited to only two nodes in the first set of nodes? For example, can your method predict a future collaboration among more than two authors?

3. If (2) can be done, then how do you score predictions which are partially correct? For example, if the actual collaboration contains 5 authors and your method predicts 4 of them correctly. How do you measure this partial success?

4. How is Node2Vec alone used for comparison? It can give you node embeddings. How do you predict links using these embeddings?

5. Overall, the proposed model performs better than the baselines. Can you connect the performance of MF-SVD to the network structure of BMS-POS and provide rationale behind its mixed performance?

Suggestions:

1. Work on the weaknesses and answers the above raised questions.

2. Authors are encouraged to check the performance of the proposed method on large scale using large datasets (given in the references above). Also, the ratio of fraction of links used to train the model and fraction of missing links need to be studied using ablation methods. Authors are encouraged to conduct this study and draw more insights to the proposed method.

References:

1. Kumar, Tarun, et al. "HPRA: Hyperedge prediction using resource allocation." Proceedings of the 12th ACM conference on web science. 2020.

2. Wan, Changlin, et al. "Principled hyperedge prediction with structural spectral features and neural networks." arXiv preprint arXiv:2106.04292 (2021).

3. Maurya, Deepak, and Balaraman Ravindran. "Hyperedge prediction using tensor eigenvalue decomposition." Journal of the Indian Institute of Science 101 (2021): 443-453.

4. Yadati, Naganand, et al. "Nhp: Neural hypergraph link prediction." Proceedings of the 29th ACM international conference on information & knowledge management. 2020.

5. Zhang, Muhan, et al. "Beyond link prediction: Predicting hyperlinks in adjacency space." Proceedings of the AAAI Conference on Artificial Intelligence. Vol. 32. No. 1. 2018.

6. Sharma, Govind, Prasanna Patil, and M. Narasimha Murty. "C3mm: clique-closure based hyperlink prediction." Proceedings of the Twenty-Ninth International Conference on International Joint Conferences on Artificial Intelligence. 2021.

6. PLOS authors have the option to publish the peer review history of their article (what does this mean?). If published, this will include your full peer review and any attached files.

Reviewer #1: No

Reviewer #2: No

---

## [Author Response · Author response to Decision Letter 0]

4 May 2024

Responses to Reviewer 1

• I strongly suggest that the authors enhance the clarity of their Abstract by clearly stating their motivation for the proposed method.

We have modified the abstract to enhance the clarity. In the revised abstract, we have explicitly stated our motivation is that we aim to propose a method "to fully learn and utilize the information extracted by FCA to improve the prediction performance".

• While the experiments conducted are sufficient, further analysis could significantly enhance the contributions and claims made in the study.

We have conducted further analysis of the main experiments and the Section "Ablation experiments". 

• It is incremental work; I still want to know the main difference and the new.

 There are three major differences between our method and previous related methods, i.e., FCA-based methods.

Difference 1: Our model can learn more information from concept lattices compared to previous FCA-based embedding methods. Previous FCA-based methods can only partly learn information from concepts, but our method can fully learn from concepts and their order relations. We have incorporated this difference into the last paragraph of “The Proposed Method” section and Figure 5.

 Additionally, we conducted two ablation experiments to demonstrate that learning more information from the concept lattice, including concepts and order relations, contributes to link prediction.

Difference 2: In our method, we employ the BERT model to learn information from the concept lattice and utilize it to make link predictions within the same model. Previous methods, however, primarily focus on generating meaningful embeddings for objects and attributes. After obtaining embeddings, they require a separate predictive model to make link predictions.

The difference between our method and previous FCA-based methods is akin to the difference between word2vec and BERT.

Difference 3: Our method can be applied to both O-O and O-A tasks, while previous FCA-based methods can only be applied to either one of them. 

Our method works in a pre-train first and fine-tune next mode, wherein we first pre-train two large models and then fine-tune them on various downstream tasks related to formal context. If we wish to conduct new tasks on a formal context where we have pre-trained the object model and attribute model, we can directly fine-tune the pre-trained model. However, previous methods require re-training the models when coming to a new task.

We have included these differences in the abstract and the end of Section "Introduction"

• In the experiment section, the latest baselines cited were published in 2020. The authors need to add some algorithms from the last three years as baselines.

We have included SBGNN, a GNN-based method, in our experiments. Proposed in 2021, it represents the state-of-the-art approach for link prediction in bipartite networks, to the best of our knowledge. 

• How was the experiment implemented, with MATLAB or Python? Will the code be publicly available?

We implemented the experiments using Python and the code is available on GitHub.

https://github.com/HYYANG9/BERT4FCA-Bipartite-Link-Predcition

• Provide more information about how the experiments were conducted, including the tools and software used, in the experiment section.

The experiments are conducted on a Windows 11 server with 64GB RAM, an AMD Ryzen 9 7900X CPU, and an NVIDIA GeForce RTX 4090 GPU.

The codes for the deep-learning part of our method are implemented with Python 3.8.18 and Pytorch 2.1.0. The code for the algorithm for extracting formal concepts is written in C++ and complied by MinGW 13.2.0.

We have added a subsection titled "Experimental environment and parameters" in the section "Experiments" to display the experimental environment and parameters.

• Provide diagram for proposed method.

We have submitted a diagram of our method in Figure 4. 

• the authors should analyse how to set the parameters of the proposed methods in the framework. Do they have the “optimal” choice?

The dimension of input embeddings of the Transformer encoder is 768, the dimension of hidden layer in Transformer encoder is 3072. 

For the BERT pre-training step, the maximum number of masked tokens used in MTP is set to 4, the dimension of embeddings is set to 768, the number of heads is set to 12, the number of Transformer encoders is set to 6. 

For the BERT fine-tuning step, the dimension of the hidden layer is set to 512. The batch size is set to 24.

Since we use a relatively complex deep-learning network, it is hard to prove whether these parameters are mathematically optimal. However, we believe they are close to the optimal choices since we have made over 20 trials and these parameters are the best among them.

We have added a subsection titled "Experimental environment and parameters" in Section "Experiments" to display the experimental environment and the parameters.

• The Literature citation is not adequate, and the related work should be discussed:

9.1 A novel link prediction algorithm for protein-protein interaction networks by attributed graph embedding

9.2. Autoencoders and their applications in machine learning: a survey

Thank you for providing us with these two papers. We found them relevant to our research and added them to our citations.

Responses to Reviewer 2

• In the definition of Concept Lattices, shouldn’t it be $B_1 \\subset \\B_2$?

The larger the object set is, the fewer attributes they will share in common. Formally, when A_1 \\subset A_2, we should have B_2 \\subset B_1. Hence, our statement in the original version of the manuscript that (A_1,B_1) < (A_2,B_2) if A_1\\subset A_2 (which mutually implies B_2\\subset B_1) is correct and do not need to be changed. 

• Is the O-O task limited to only two nodes in the first set of nodes? For example, can your method predict a future collaboration among more than two authors?

In the O-O task, our method (as well as all previous methods we implemented) can predict future collaboration among any number of nodes. However, due to limitations of computational resources, in this research, we only include the collaborations among up to 5 nodes in our training and test cases.

We have also included the above statements (in a more formal and detailed way) in a new subsection "Generation of Labeled Training and Test Samples" in Section "Experiments" so that our readers can also be clear about the form of training and test data.

• If (2) can be done, then how do you score predictions which are partially correct? For example, if the actual collaboration contains 5 authors and your method predicts 4 of them correctly. How do you measure this partial success?

As mentioned above, we have added a new subsection "Generation of Labeled Training and Test Samples" in Section "Experiments". In this section, we describe the rules for generating test samples. With such rule, if a new collaboration contains 5 authors, then all the subsets of these 5 authors are also included in the test set. Hence, suppose that the new collaboration is (o1, o2, o3, o4, o5), then even if a method fails to predict the sample (o1, o2, o3, o4, o5), it may still gain true positives if it can correctly predict the sample (o1, o2, o3, o4), (o1, o2, o3, o5), ..., (o1,o2,o3), ... etc. This is how we measure partial prediction success.

• How is Node2Vec alone used for comparison? It can give you node embeddings. How do you predict links using these embeddings?

After using node2vec to obtain the embeddings vectors for each node, a.k.a., each object and attribute, we conduct a downstream classification task with the embedding vectors to make predictions. Specifically, we first generate labeled training and test samples as introduced in the newly added subsection "Generation of Labeled Training and Test Samples " in Section "Experiments". Then, for each sample, its feature is set to the concatenation of the embeddings vectors of the group of objects (for the O-O task) or the object-attribute pair (for the O-A task). The processed samples are fed into a Logistic Regression classifier to train a model which can classify existing and non-existing links, which can thus be used for link prediction.

We have included the above description in Subsections "The Experiment of the O-O Task" and "The Experiment of the O-A Task" of Section "Experiments".

5. Overall, the proposed model performs better than the baselines. Can you connect the performance of MF-SVD to the network structure of BMS-POS and provide rationale behind its mixed performance?

We have added the analysis on why MF-SVD has a mixed performance on BMS-POS into Subsection "The Experiment of the O-A Task" of Section "Experiments". The added analysis is as follows.

MF-SVD exhibits higher AUC scores than other methods across all three datasets but with very low AUPR scores. According to previous research, the AUPR score gives more weight to the positive samples. A low AUPR score indicates that the model's high prediction scores do not correlate well with being in the positive class, suggesting that the model has difficulty achieving high precision. Since in link prediction, we focus on predicting the generation of new links, which is predicting the positive samples but not the negative samples, so if a model exhibits a low AUPR, it is considered to have low performance in link prediction.

The low performance of MF-SVD may suggest that the underlying latent features of the networks and the relationships between nodes are intricate and challenging for MF to capture effectively. Additionally, since the BMS-POS dataset is extracted from a large dataset, maybe the relationships between nodes in this network are not consistent and stable, resulting in low prediction performance of MF.

Also, after conducting experiments on the two extra datasets, we have found that Node2Vec also has a mixed performance - it performs well on the iJO1366 dataset but performs bad on the Review dataset. We also add the analysis into the same subsection. The added analysis is as follows.

In the iJO1366 dataset, Node2Vec exhibits higher AUPR and AUC scores than our method. However, its scores in the Review dataset is far lower than our method, indicating that the performance of Node2Vec varies greatly across different datasets. According to previous research, the sparser the network is, the better the process can capture the features of the network. Hence, it is expected to work better on sparse networks. Since iJO1366 is the most sparse data set, Node2Vec achieved to gain a better performance, but for relative dense networks, its performances become notably lower. Hence, we consider our method to be more practical than Node2Vec thanks to its overall good performance in all metrics across all datasets.

Weaknesses:

• Unclear novelty: The exact novelty of the proposal and what exactly derives superior performance of the proposed method is not clearly laid out.

The main novelties are as follows:

• We employ a BERT-like transformer encoder network to learn information provided by FCA and utilize it for two link prediction tasks within the same model. However, previous methods primarily focus on generating meaningful embeddings for objects and attributes, and the transformer encoder network is not concerned, to the best of our knowledge. 

• Our method can fully learn information from concept lattices, including concepts and their order relations, while previous FCA-based methods can only partly learn information from concepts.

We have modified the abstract and Section "Introduction" to enhance the clarity of the novelty.

The reasons for the superior performance of our method are as follows:

• Our model can learn more information from the concept lattice, and we conduct two ablation experiments to demonstrate that it is beneficial for bipartite link prediction.

• Our model can directly extract useful information for link prediction from the formal context (which is equivalent to a bipartite network), in addition to learning from the concept lattice. During the fine-tuning step, we train the model with samples generated from the formal context, which enables the model to learn directly from the formal context. The second ablation experiment demonstrates that when solely relying on bipartite networks, our model can also outperformance other methods. 

We have included this analysis in the last paragraph of Section "The Proposed Method" and Figure 5. Additionally, we have made some further analysis in the "Experiment" section to dive into the superior performance of our method.

• Missing prominent baselines: Notably, several relevant baselines from related disciplines, such as hyperedge prediction in hypergraphs, have been overlooked. These baselines could provide valuable insights and should be included for a comprehensive evaluation (follow the references at end).

 We would like to thank you for providing us the information on the field of hyperedge prediction. After surveying the listed papers, we have confirmed that the task of hyperedge prediction is similar to the O-O task in bipartite link prediction, and thus these methods can be directly applied to our task as well. 

 We have added a relatively new hyperedge prediction method, HPRA, as well as two classic hyperedge prediction methods - Common Neighbors and Katz as baselines. New statistics and analysis are added to Section "Experiments". Also, as suggested by the other reviewer, we have also added a new baseline for the O-A task. We believe that the comprehensiveness of evaluation is improved now.

3. Limited Dataset and Ablation Study: The datasets considered are very few in numbers and very small in the size. Also, there is no ablation study which makes it hard to understand how the proposed method works and leads to these results.

 We have added two more datasets used in previous research into our experiments. We believe the results on these two datasets can provide a better proof of the advantage of our method.

 We cannot further increase the number of the datasets because our method is not designed to make predictions on large datasets. Our method, as well as all FCA-based methods, requires to extract formal concepts from the formal context corresponding to the input bipartite network. If the network is too large, i.e., it has too many nodes or edges, then its corresponding formal context will likely have an excessive number of formal concepts, because the number of formal concepts usually increases exponentially as the number of nodes and edges increases. Since the lower bound of time complexity of extracting concepts is O(C), where C is the number of formal concepts. Extracting all concepts from a large dataset may take several days. 

We consider this as a limitation of our method. We have mentioned this limitation in Section "Conclusion and Future Work" in the revised version of the manuscript. 

We have already conducted ablation experiments to study whether our model has indeed captured the neighboring relations, and whether the captured neighboring relations contribute to the good performance of predictions. The experimental results and discussion were in subsection "Supplementary Experiments" of Section "Experiments" in the manuscript before revision. To make it clearer, in the revised version of the manuscript, we changed the title of the subsection to "Ablation Experiments".

Suggestions:

1. Work on the weaknesses and answers the above raised questions.

2. Authors are encouraged to check the performance of the proposed method on large scale using large datasets (given in the references above). Also, the ratio of fraction of links used to train the model and fraction of missing links need to be studied using ablation methods. Authors are encouraged to conduct this study and draw more insights to the proposed method.

Use larger datasets:

As mentioned above, our method is not desigend for large datasets. To get such a high performance it requires extra execution time, and therefore, we are unable to apply our method to the datasets in the references you provided, as it will cause a lot of 

---

## [Decision Letter · Decision Letter 1]

21 May 2024

BERT4FCA: A Method for Bipartite Link Prediction using Formal Concept Analysis and BERT

PONE-D-24-07133R1

Dear Dr. Yang,

We’re pleased to inform you that your manuscript has been judged scientifically suitable for publication and will be formally accepted for publication once it meets all outstanding technical requirements.

Kind regards,

Karthik Raman, Ph.D.

Academic Editor

PLOS ONE

Additional Editor Comments (optional):

I am happy to recommend the acceptance of this manuscript given that all the reviewer comments have been satisfactorily addressed.

Reviewers' comments:

Reviewer's Responses to Questions

**Comments to the Author**

1. If the authors have adequately addressed your comments raised in a previous round of review and you feel that this manuscript is now acceptable for publication, you may indicate that here to bypass the “Comments to the Author” section, enter your conflict of interest statement in the “Confidential to Editor” section, and submit your "Accept" recommendation.

Reviewer #1: (No Response)

Reviewer #2: All comments have been addressed

2. Is the manuscript technically sound, and do the data support the conclusions?

Reviewer #1: Yes

Reviewer #2: Yes

3. Has the statistical analysis been performed appropriately and rigorously? 

Reviewer #1: Yes

Reviewer #2: Yes

4. Have the authors made all data underlying the findings in their manuscript fully available?

Reviewer #1: (No Response)

Reviewer #2: (No Response)

5. Is the manuscript presented in an intelligible fashion and written in standard English?

Reviewer #1: Yes

Reviewer #2: Yes

6. Review Comments to the Author

Reviewer #1: The conclusions of the manuscript, which include all the corrections made by the authors, are properly explained and well organised throughout the entire manuscript.

Reviewer #2: Thank you for addressing my comments. My major concerns have been resolved. However, I have two minor suggestions before the manuscript is published:

Evaluation of the Proposed Method: I understand that you include all possible subsets of the groups in the test set. However, this approach has some drawbacks. If the entire group is predicted, all its subsets that are not predicted will result in a lower score. Additionally, this method gives equal weight to predicting the entire group and predicting its subset, which may not be a valid approach. Please consider addressing this issue in the experiment section.

Code and Data Availability: If possible, please make the code and data publicly available so the community can benefit from your proposed method.

Overall, excellent work! Best of luck!

7. PLOS authors have the option to publish the peer review history of their article (what does this mean?). If published, this will include your full peer review and any attached files.

Reviewer #1: No

Reviewer #2: No

---

## [Editor Report · Acceptance letter]

23 May 2024

PONE-D-24-07133R1 

PLOS ONE

Dear Dr. Yang, 

I'm pleased to inform you that your manuscript has been deemed suitable for publication in PLOS ONE. Congratulations! Your manuscript is now being handed over to our production team.

Kind regards, 

on behalf of

Dr. Karthik Raman 

Academic Editor

PLOS ONE